# Short-Term Acceptability of Ready-to-Use Therapeutic Foods in Two Provinces of Lao People’s Democratic Republic

**DOI:** 10.3390/nu15173847

**Published:** 2023-09-03

**Authors:** Iacopo Aiello, Sengchanh Kounnavong, Hari Vinathan, Khamseng Philavong, Chanthaly Luangphaxay, Somphone Soukhavong, Janneke Blomberg, Frank T. Wieringa

**Affiliations:** 1Faculty of Medicine, Aix-Marseille University, AP-HM, 13385 Marseille, France; aiello.iacopo@gmail.com; 2Food, Nutrition, Health, UMR QualiSud, French National Research Institute for Sustainable Development (IRD), 34394 Montpellier, France; 3Health and Nutrition Section, UNICEF, Vientiane, Laos; hvinathan@unicef.org (H.V.); jblomberg@unicef.org (J.B.); 4Lao Tropical and Public Health Institute, Ban Kaognot, Sisattanack District, Vientiane, Laos; skounnavong@gmail.com (S.K.); chanthaly3@hotmail.com (C.L.); 5Centre of Nutrition (CoN), Ministry of Health (MoH), Ban Xiengda, Vientiane, Laos; khamseng_p@hotmail.com; 6Faculty of Public Health, University of Health Sciences, Vientiane, Laos; s.somphone213@gmail.com; 7UMR QualiSud, CIRAD, University of Montpellier, SupAgro, IRD, University of Avignon, University of Reunion, 34394 Montpellier, France

**Keywords:** severe acute malnutrition, RUTF, acceptability, Laos, children, wasting

## Abstract

Background: In Lao PDR, acute malnutrition remains a public health problem, with around 9% of children under 5 being affected. Outpatient treatment of severe acute malnutrition requires ready-to-use therapeutic foods (RUTFs), but concerns have been raised about the acceptability of globally available products. Culturally acceptable RUTFs could be locally developed, but data are lacking on RUTF preferences in Lao PDR. Methods: In a crossover-designed study, four different RUTFs were tested for short-term acceptability and organoleptic qualities (two globally available: peanut-based, which is the current standard, and wheat–milk-based RUTFs; two regionally produced: a mung-bean-based and a fish–rice-based RUTF). Organoleptic properties were evaluated by 83 caretaker–child pair participants through a taste test and a 30 min consumption test. Short-term acceptability was assessed through a 3-day intake test. The study sites were in Phongsaly (North Laos) and Attapeu (South Laos). Focus group discussions were conducted at the beginning and the end of the study. Results: The mung bean RUTF was the favorite among caretakers, with an acceptability percentage of 96.2%, and scored better (*p*-value < 0.05) for all organoleptic variables than the other three RUTFs. Overall, 3 days after receiving take-home rations, the mean percentage of consumption was above 80% for all the RUTFs, with the mung bean product being the most consumed. Conclusions: The regionally produced mung bean RUTF was the most accepted, whereas the other regionally produced fish-based RUTF was the least accepted, showing the complexity of finding culturally acceptable solutions to fight malnutrition. For Lao PDR, a mung-bean-based RUTF seems the way forward, even if the current standard peanut-based RUTF appeared to be acceptable, albeit not the most preferred.

## 1. Introduction

Malnutrition, including undernutrition, micronutrient deficiencies, and overnutrition, continues to be a major public health problem in most low- and middle-income countries (LMIC), particularly in the South-East Asia Region [1]. Acute malnutrition, or wasting, is classified according to its severity as either moderate acute malnutrition (MAM) or severe acute malnutrition (SAM). SAM in children below 59 months of age is defined as a weight-for-height/length z-score (WHZ/WLZ) of more than three standard deviations below the mean and/or a mid-upper arm circumference (MUAC) of less than 115 mm, or the presence of bilateral edema [2].

Severe acute malnutrition affects nearly 20 million preschool-aged children, and it represents a significant factor in approximately one-third to one-half of the nearly 8 million deaths of children under 5 years of age worldwide [3,4,5]. Indeed, wasting has been shown to be a strong risk factor for mortality: children with severe wasting have more than 11-fold higher mortality rates than children with WHZ ≥ −1 [6]. Moreover, SAM in early childhood can lead to negative health consequences later in life [7].

Even though Lao PDR has made important steps in economic development over the last few decades, it still experiences moderate-to-serious hunger levels according to the 2021 Global Hunger Index [8]. In fact, UNICEF estimates that there are at least 60,000 children suffering from severe acute malnutrition annually in Lao PDR, or almost 3% of all children < 5 years of age [3,9]. However, there are remarkable provincial differences within the country (SAM prevalence ranging from 0.7% to more than 8%) [9,10]. It is worrisome that, from 2006 to 2017, a worsening trend in acute malnutrition prevalence in Lao PDR is evident [9,11].

For SAM without complications, or when complications have been addressed already, home-based or community-based treatment is recommended, including the provision of ready-to-use therapeutic foods (RUTFs). Indeed, home- or community-based treatment of children with SAM has proven to be as, or even more, successful as hospital-based treatment [12], and UNICEF promotes the Integrated Management of Acute Malnutrition (IMAM), which includes community-based management of acute malnutrition (CMAM).

An RUTF is an essential part of the community-based treatment for severe acute malnutrition, in that it provides all the micro- and macro-nutrients required for recovery [13]. As its name implies, it is ready-to-use and can easily be consumed at home because it neither requires any preparation nor cooking before consumption, and depending on the age, it can be consumed also under minimal supervision until adequate weight has been gained. An average full course of treatment for an infant or a child amounts to around 10/15 kg of RUTF over a six-to-eight-week period. Thus, by exclusively consuming small portions of an RUTF five to seven times a day, a severely malnourished child can achieve sufficient nutrient intake for full recovery in a few weeks [13]. Although proven to be a successful and effective intervention for treating SAM, the use of RUTFs is not without drawbacks, as, for example, the acceptability of RUTFs is a concern. The term acceptability not only refers to the taste of an RUTF but also to other organoleptic qualities (consistency, color, palatability, smell, and appearance) and to the more complex cultural and social relations occurring at different levels (intra- and inter-familiar), between implementers, health medical staff, villagers, caretakers, and so on [14]. To address these issues, where local possibilities exist to create sustained improved nutrition for children by means of local, more affordable, more acceptable, and effective RUTF development and deployment, WHO and UNICEF have declared to be fully supportive of these efforts [13]. UNICEF and IRD (the French National Research Institute for Sustainable Development) have been collaborating on the development of locally produced ready-to-use foods (both therapeutic and supplementary) in South-East Asia (Vietnam [15,16,17] and Cambodia [18,19,20]) since 2009.

The Ministry of Health of Lao PDR expressed concern that the currently used RUTFs are potentially not acceptable. Adapting RUTFs to local tastes and preferences using regionally available ingredients (for example, in the case of South-East Asia: mung beans, soybeans, rice, coconut, etc.) could result in higher product acceptability. Moreover, local manufacturing of RUTFs makes countries more independent from importing RUTFs [21] and contributes to local food systems. The development of a potential novel, locally produced, and culturally acceptable RUTF would also respond to the Lao PDR programmatic need to scale up the community treatment of SAM to accelerate the reduction in child malnutrition and prevent further worsening trends [22].

However, before further exploring local production of an RUTF, more information is needed on the organoleptic preferences for an RUTF to be used for treating children with SAM in Lao PDR. Thus, the current study was conducted to describe preferences for existing, non-Lao RUTFs, among caregivers and children, to understand the acceptability of the currently used product and to potentially facilitate the future development of a Lao RUTF.

## 2. Materials and Methods

The study took place in January and February 2022 in two provinces in Lao PDR: one in the north (Phongsaly, Muang Mai village) and one in the south (Attapeu, Sanamxai village). The provinces were selected to consider feasibility, the incidence of acute malnutrition, and potential differences in organoleptic preferences between different regions of Lao PDR.

A total of around 80 child–caretaker pairs were involved (40 pairs in each province set as recruitment target). Inclusion criteria comprised children aged 12–59 months with weight for height/length between −3 and 0 Z-score, and a mid-upper arm circumference (MUAC) > 11.5 cm, accompanied by their caretakers and not planning to move during the study period. Exclusion criteria included no solid food intake for less than three months or known specific food allergies or intolerance (e.g., soy, nuts, or dairy allergy), and any child found with SAM (who indeed was to be referred to the health system for appropriate treatment as required, as the study is an acceptability study, and not a treatment for SAM) was not recruited for ethical reasons. Recruitment was carried out with the convenience sampling technique, with caregivers informed and invited to participate by the health center. Child–caretaker pairs who gave oral and written consent were included in the study.

Data were collected by staff from the Lao National Nutrition Centre and Lao Tropical and Public Health Institute (TPHI) who had received a 2-day training session on anthropometric measures and the questionnaires.

Height was measured to the nearest 0.10 cm using a height board, weight was assessed to the nearest 0.10 kg, using a flat digital weighing scale, which was calibrated regularly. MUAC was measured using a standardized MUAC strip to the nearest 0.10 cm. All measurements were taken in duplicate. If the difference between the two measurements was more than 0.2 cm or 0.2 kg, the measurements were repeated. Z-scores (WAZ, HAZ, WHZ) were calculated using the 2007 World Health Organization (WHO) growth reference standard and using the R package “anthro”, provided by WHO [23].

Four different types of ready-to-use therapeutic foods were tested: BP-100 (a bar), HEBI (compressed bean cake), eeZeePaste (paste), and Nutrix (a fish-paste-filled wafer) (Table 1).

The overall study consisted of 2 different sub-studies, with the first study evaluating the organoleptic qualities of the RUTF, and the second being a consumption trial, measuring the 3-day consumption of the RUTF, which were provided in a cross-over-designed take-home ration study. To prevent caregivers and children from receiving the RUTF in a fixed order, caregiver–child pairs were randomized to one of 10 different sequences of receiving the RUTF (4 caregiver–child pairs per sequence in each province). This random sequence was used for both testing the organoleptic qualities as well as for allocating the RUTF in the cross-over-designed take-home ration study.

Caretakers were asked to taste each of the four RUTFs to assess hedonic organoleptic qualities and differences among the RUTFs. Caregivers were asked to score color, smell, taste, look, texture, and overall quality and indicate their preferred and least favorite RUTF. Each caretaker completed the questionnaire independently without interference from the research team or the other participants. Each caregiver was presented with a white plastic plate with each of the RUTFs. The caretakers were asked to taste each product in random order and score the organoleptic qualities of each product. The scoring system included 3 levels of rating—bad, neutral, or good—which were indicated by corresponding pictorial emoticons/smiley faces. The caretakers were given a cup of water to rinse and drink after each tasting. The products were presented to the caretakers in random order, through the sequence random list prepared before the start of the study. After the tasting session by caretakers, a short focus group discussion (FGD1) was held.

After FDG1, a chronometric taste test was performed with children with the first RUTF from the sequence random list. This chronometric taste test was conducted to assess the amount of RUTF consumed in 15 and 30 min, as a means to evaluate their acceptability among children. The child was free to eat the presented RUTF, and after 15 min (T15), the remaining amount of product was weighed to determine the amount consumed by the child. The remaining product was returned to the child to continue eating until 30 min (T30) when again the remaining product was collected and weighed. This appetite test, adapted from the UNICEF appetite test [24], is a standard test before commencing community-based treatment of SAM. We defined the quantity of RUTF eaten after 30 min to be “acceptable” if at least 50% of the initial amount was consumed [17]. The chronometric taste test was repeated on each day the caregiver and child came back to collect the next batch of RUTF during the cross-over take-home ration study, again according to the random sequence list (days 4, 8, and 12; see Figure 1 below).

Hence, to assess short-term acceptability, each child received each RUTF product for 3 consecutive days but in different orders. Each caretaker was instructed on the use of the product first and asked to give one sachet (±100 g) to the child every day for 3 days (one sachet per day). After this period, one “wash-out” day was inserted. The process was repeated, with other wash-out days after 3 days of intake, until all children had tested the 4 products. Caregivers were provided 290–300 g of eeZeePaste, HEBI, and BP-100 for each of the 3 days and 360 g for Nutrix due to the different forms of packaging of the latter. Caregivers were asked to bring back all sachets with RUTF on the wash-out day. The total amount of eaten RUTF after 3 days was calculated from the amount of RUTF returned.

On the final wash-out day, a long FGD (FGD2) was performed to gather information on the different RUTFs and on the subjective experience throughout the take-home ration study. Qualitative data collected during the focus group discussions (conducted in the local language) were recorded, transcribed, and translated; coded into phrases, sentences, or paragraphs; and subjected to qualitative (thematic) analysis.

In Phongsaly, due to the high diversity of ethnic groups in this northern province of Lao PDR (Hmong), the FGDs were conducted in the local Hmong language, and translated into the Lao language first and, thereafter, English. Figure 1 provides a graphic illustration of the study implementation.

Quantitative analyses were performed both for all participants together and separately for the two provinces. After considering assumptions and conditions, appropriate statistical methods were selected for data analysis, accordingly: non-parametric tests were used when the conditions of application were not met and Bonferroni correction was used when several tests were being performed simultaneously to correct type I error. Qualitative methods guided the project, and the quantitative ones provided a supporting role in the procedures to confirm, cross-validate, or corroborate qualitative findings. Quantitative data were analyzed using R statistical software (R for R 4.0.2 GUI 1.72 Catalina build) and Microsoft Excel spreadsheet for Mac 16.76, and verbatim accounts were subjected to content and thematic analyses using a framework approach.

The study protocol received approval from the National Ethic Committee for Health Research of the Ministry of Health, Lao PDR.

## 3. Results

The study included a total of 83 caretaker–child pair participants: 41 in Phongsaly province and 42 in Attapeu province. All caretakers participated in the hedonic organoleptic test, but in Attapeu, two children did not take part in the chronometric intake test and the cross-over take-home ration study. In the same province, three caretaker–child pairs did not come back at the third round of the cross-over take-home ration study (day 12), and one pair did not come back at the fourth and last round of the cross-over take-home ration study and, thus, did not participate in FGD2.

In Phongsaly, 36% of caregivers were males, whereas in Attapeu, only 2% were males (χ² test *p*-value < 0.05). Among children, the mean weight-for-height Z-score was significantly lower in Phongsaly, but the prevalence of MAM was similar. Otherwise, no statistically significant differences existed between the two provinces (*p*-value > 0.05; see Table 2 for details about children’s characteristics).

### 3.1. Caretakers’ Organoleptic Qualities

Mean organoleptic scores for each product, as given by the caregivers, are shown in Table 3.

The mung bean bar (HEBI) received the highest overall score (17.33 out of 18), followed by BP-100 (16.51) and eeZeePaste (15.54), while Nutrix received the lowest overall score (13.56 out of 18). No statistically significant differences existed between Phongsaly and Attapeu, considering the six different variables, except for Nutrix, which was less liked in Attapeu compared to Phongsaly, as far as it concerns color, smell, taste, and texture variables (Welch’s *t*-test and *t*-test *p*-value < 0.05), and for eeZeePaste, the smell of which appeared to be less liked in Phongsaly than Attapeu (Welch’s *t*-test *p*-value < 0.05). HEBI scored better for the six organoleptic characteristics compared to Nutrix (Welch *t*-test *p*-value < 0.05; see Figure 2), and HEBI was also selected as the most favorite RUTF by more than half of the caretakers (53%, both provinces combined), while Nutrix scored the worst for five out of 6=six organoleptic characteristics. Nutrix was selected as the least favorite RUTF by almost 70% of caretakers (both provinces combined, Fisher’s exact test *p*-value < 0.05).

### 3.2. Hedonic Chronometric Test in Children

At 15 min, the only RUTF that had been consumed >50% of the provided amount was eeZeePaste (50.4%), followed by HEBI (44.6%), BP-100 (46.6%), and Nutrix, which was the least consumed after 15 min (34.9%,). Because of the very wide standard deviations, the only statistically significant differences found at 15 min were between BP-100 and eeZeePaste and between Nutrix and eeZeePaste (ANOVA with Bonferroni correction *p*-value < 0.0083).

After 30 min, children had consumed >50% of all the RUTFs tested. On average, children consumed between 51.1% (Nutrix) and 59.5% (eeZeePaste) of the provided RUTF (ANOVA *p*-value > 0.05). As for passing the appetite test (i.e., having eaten more than half of an RUTF in 30 min), the percentage of children who ate more than 50% of each RUTF after 30 min ranged from 55% for HEBI and eeZeePaste to 52% for Nutrix and only 40% for BP-100 (χ² test *p*-value > 0.05).

Consumption of the four RUTFs during the 3 days of take-home rations (THRs) was above 80% for all four tested RUTFs (see Table 4). HEBI and BP-100 consumption was significantly higher than for Nutrix (Welch’s ANOVA with Bonferroni correction, *p*-value < 0.0083), but no statistically significant difference was found between the other three RUTFs (HEBI, BP-100, and eeZeePaste). While over 90% of HEBI (93.4%), BP100 (92.4%), and eeZeePaste (92.3%) were consumed, only 81.8% of the provided Nutrix was consumed. There was a significant increase in the average amount of RUTF consumed from day 1 to 3 to day 13 to 15, regardless of the RUTF.

### 3.3. Focus Group Discussions

In Phongsaly, FDG1 consisted of four different interviews, while in Attapeu, eight interviews were conducted with smaller groups (from 2 to 14 child–caretaker pairs each). In Attapeu, in the south of Lao PDR, the majority of groups explicitly referred to malnutrition as a status of both loss of appetite and anorexia, and also to physical and mental exhaustion and decay (“*Malnutrition is always sick, thin, and anorexia. They don’t want to eat meals. Small children, often unwell, go to hospital often*” and “*Small child, short, child often gets sick, inadequate food*”). In northern Phongsaly, caretakers focused more on the bodily consequences of malnutrition, rather than on the mental and psychophysical impact on children (“*Malnutrition is weakness, big belly, and small legs, hair color, and face have change. Children are thin*”). Also, they were quite aware of the association between the lack of sufficient food intake and malnutrition (“*No have enough food for children, don’t eat enough food, no have food to eat can be malnutrition*”). In both provinces, malnutrition was regarded as a problem: “*That is exactly a big problem in this village and still have a lot of malnutrition in children. Mostly under 5 years. We don’t have enough money for buy the meals that has more nutrition for cooking*” (Attapeu) or “*In our village have malnutrition problem because there is not enough food, it bases on nature food. Because no have market, no have any condition to buy, no have land to planting, some trees have no fruits, we have small garden in the field but there is only lettuce. Normally we go to the forest to find food*.” (Phongsaly). Therefore, a condition of lack and shortage of food is evoked, particularly in Phongsaly province, where it appears that caretakers often need to go and find themselves food directly in the forest or in the wild, for example, also because apparently no markets are available nearby. In Attapeu province, caretakers raised the problem of not having enough money to buy and afford food, more than the scarcity of food itself.

Concerning feeding practices, in Phongsaly province, caretakers differentiated between feeding practices before and after 6 months (“*Over 6 months, feed children by rice, vegetables, meat, chicken, duck, squirrel, or wild animals*”). During the first six months, they stick more to breast feeding but not exclusively, with also rice soup and crackers included in the diet. In Attapeu province, no difference in feeding habits between less and more than 6 months was mentioned (only one group stated that “*babies like to breastfeed*”), but more diverse food types were cited, including fruits, fish, tubers, soy milk, and eggs. Interestingly, in Phongsaly and Attapeu, people were unaware of what to do exactly when a child was perceived to be malnourished, other than feed as usual (“*give rice, don’t know how to do*”, Phongsaly; “*don’t know anything to do*”, to “*give child more to eat: fish, vegetables, fruit, milk, give supplements*”, to “*go to the doctor*”, Attapeu).

Lastly, the issue of whether they would give specific food or make special meals in the case of malnutrition was specifically and explicitly addressed. In Phongsaly, they all agreed on the fact that no special food was available to them (“*eat what we have*”), and that they “*don’t know how to make special food*”. There was one group in Attapeu that declared not to have enough money to afford new and different foods in the case of malnutrition.

During FDG2, caretakers confirmed their understanding and knowledge of malnutrition as a condition that affects the body, mind, and might include loss of appetite. Caretakers’ impression of the RUTF provided was that most children liked the products, although in Attapeu there were “*likes and dislikes*”. Appreciation of an RUTF was expressed as “*like a snack*”, and in caretakers’ opinions, children liked one product when children “*ate it all*”, “*finished it since day one*”, and were “*happy to eat*”, while refusal or dislike was expressed as “*cry, vomit, yell, and throw away*” or “*have to force to eat, they eat a little bit, eat slowly, and throw*”. The majority of all groups agreed on the fact that Nutrix was the worst one among the RUTF (“*has a bit of a fishy smell*”; “*Nutrix taste is not delicious, and texture is salty a little bit*”) and that the worst side of it was the “*fishy odor*”. As for BP-100, most groups agreed on the excessive salinity and insufficient sweet taste. EeZeePaste was suggested to be white or lighter in color and to also be less savory in taste. The most appreciated RUTF by caretakers appeared to be HEBI. In Attapeu, one group noted that Nutrix and BP-100 were the RUTF most refused. In addition, in Attapeu, there were two groups reporting their children “*had diarrhea*”. Unfortunately, it is difficult to say to which extent, following which product, and under which baseline health conditions, these children experienced these transient negative outcomes, reported late. In Phongsaly, no such events were recorded (but vomiting).

Finally, as far as it concerns the evaluation of the RUTF packaging, there was a widespread suggestion on adding “*pictures*” to it, to make it more attractive. Two groups in Phongsaly proposed to add “*multiple colors*” to ameliorate the packaging and attractivity of the product (“*green, blue, and yellow*”).

## 4. Discussion

In the present study, we aimed to compare the organoleptic qualities and acceptability of two regionally produced RUTFs and two globally available RUTFs, with the long-term objective of developing an RUTF based on local organoleptic preferences. Caretakers consistently agreed that HEBI, an RUTF based on mung bean and produced in Vietnam, was the most preferred RUTF, while Nutrix, produced in Cambodia and containing fish, was the least appreciated. Worrisome is that 1/3 of caregivers in Phongsaly indicated that the peanut-based RUTF was their least favorite RUTF, as this RUTF is the current standard in Lao PDR. Interestingly, children showed a more nuanced preference, especially with eeZeePaste (peanut-based) and HEBI being appreciated when regarding the amount of RUTF consumed and, again, Nutrix being the least appreciated. At 30 min, children appeared to consume more eeZeePaste than HEBI (not statistically significant), while the opposite was true for the 3 days of take-home rations (THRs). But, this apparent discrepancy between the acceptability outcome at 30 min and 3 days could be easily explained by the fact that children’s taste and acceptability could adapt with time, with ongoing consumption and exposure to the same product (a phenomenon known as “food neophobia”, i.e., the reluctance to eat new or unfamiliar foods), leading to an increase in consumption [25,26]. Another explanation could be that during the 3 days of THR, other household members or siblings might have consumed some of the RUTFs. In any case, Nutrix appeared to be the least favorite RUTF for both children and caretakers and for all the explored variables and components. Although rice is a staple food in Lao PDR, and Nutrix’s main ingredients include rice and fish, the reason for rejecting this RUTF appeared to be the fish component and its perceived strong taste and strong odor (probably very far from the usual taste of food for children).

Since the acceptability of an RUTF by a child is a multifaceted process that includes also cultural and social components, acceptance by a caretaker is of importance for final acceptance by a child, not least as the child will always be dependent upon caretakers for the intake and consumption of the RUTF during the treatment of acute malnutrition. Therefore, if the caretaker’s acceptance is high, it is most likely that the child’s will be high too. This is the reason why, in the present study, importance was also given to the acceptability and organoleptic preferences of caretakers.

Our findings are in accordance with a previous study conducted in Vietnam, which showed that HEBI was well accepted [15]. Furthermore, HEBI was manufactured as a small bar (compressed cube), rather than a paste, which appeared to be a more attractive factor for consumption by children. Our findings also partly confirm the results from a recent trial in Indonesia, which showed that locally produced RUTFs can have higher acceptance and consumption than a standard, peanut-based RUTF [27]. Interestingly, in the trial in Indonesia, the most consumed RUTF was also mung-bean-based, as in our trial, although a second mung-bean-based RUTF did not show higher consumption than the control, a peanut-based RUTF [27].

During the FGDs in Phongsaly province, it has emerged multiple times that often, there was not enough food to properly feed children, suggesting a high prevalence of food insecurity. Indeed, this is confirmed by the socio-economic, educational, and WASH indicators for this province, ranking it among the least developed in the country [9,28]. Barter trade is still a common practice in Phongsaly province [29], and this could also explain why, in Attapeu, caretakers referred more often to the lack of money to afford nutritious food rather than the lack of food itself. In both provinces, caretakers did not really know what to do in the case of a malnourished child, as far as it concerns the diet. One reason for this might be that the IMAM program in Lao PDR is still in a very early phase of implementation, meaning that it is not yet in place in many provinces. It is hoped that as the quality and access to IMAM services steadily improve over the coming years, efforts for improving community awareness of acute malnutrition are also implemented (including enhanced efforts for early case finding and referrals).

During the FGDs after the end of the study, some caretakers mentioned the occurrence of vomiting and diarrhea during the study. Diarrhea during treatment for SAM with RUTFs has been observed before and can be explained by the fact that these RUTFs are very rich in fats and that some children with SAM have fat intolerance, leading to (transient) diarrhea [30]. Also, there might have been a mistranslation or misinterpretation of the word “vomit” for “regurgitation”, a phenomenon that is common in children and not a phenomenon to specifically worry about.

In caretakers’ opinions, the primary factors determining children’s acceptability of an RUTF are smell and appearance, which make an RUTF appealing, thus possibly making them willing to eat it. This is the reason why HEBI was the most preferred one (nice smell, nice taste, and nice look), and Nutrix the least favorite one (mostly due to its fishy odor). Nutrix was only appreciated because it looked like a snack (a wafer), but its odor was so strong that it annulled this positive characteristic. EeZeePaste was, overall, well appreciated by caretakers and noticeably consumed by children, but the not-so-appealing color was pointed out, and a suggestion to make it whiter or lighter was given multiple times. Interestingly, excessive salinity was raised as a negative point quite a few times for three out of the four tested RUTFs, suggesting ways to improve the overall acceptability of RUTFs, such as lowering the perceived saltiness, for example, through a different chemical formulation of the vitamin–mineral premix.

This study also has some limitations that should be taken into consideration when applying its results. First, only two provinces were included from among Lao PDR’s 18 provinces, which have very diverse cultures and food preferences. Therefore, although this study aimed at studying RUTF acceptability in Lao PDR, caution should be applied when generalizing the results to a broader Laotian population that consists of numerous and quite distinct and diverse ethnic groups (47 different ethnolinguistic groups) [28,31]. Therefore, a more extensive study (acceptability of a new Lao RUTF prototype), involving more provinces in the country, would be advisable.

Second, the FGDs had to be translated into English, either directly from Lao or in Phongsaly first from the local Hmong language to Lao and then into English, potentially leading to the introduction of errors. Translating FGD data requires way more than fluency and knowledge of the languages, as different contextual, cultural, and social factors might influence the highly complex process of translation. To overcome this, in-depth training and active outreach sensitization were carefully carried out with all the assistants and translators involved in the project.

Third, by the nature of the research itself, this acceptability study involved a relatively short period of exposure to the RUTF, as the study provided four different RUTFs over a short period of time (3 days), while an actual treatment intervention requires the intake of the same product for longer periods of time (typically 6 to 8 weeks). Whether acceptance of the product would remain at the same levels during a treatment intervention can only be assessed through for example an effectiveness study. The earlier-mentioned study in Indonesia showed a sharp reduction in RUTF consumption after the first 4 weeks [27].

Finally, the roles of other stakeholders, in particular, health center workers and community leaders, were not assessed in this acceptability study; yet, their role remains crucial in the acceptability of a product for the treatment of acute malnutrition. Increasing their involvement could be part of an effectiveness study as well. Moreover, sample selection was limited to individuals who spontaneously enrolled through their respective village health centers after a call for interest and who were available to participate in the study on the days scheduled. The specificity of the sample could, thus, limit the generalizability of study results to a broader population and, in particular, to populations affected by severe acute malnutrition.

## 5. Conclusions

Even though all four tested RUTF products were proven to be generally acceptable in the population involved in this study, there was a clear preference for the mung-bean-based RUTF produced in Vietnam. Therefore, a novel, locally produced RUTF could be one with organoleptic characteristics similar to the RUTF from Vietnam. This would also respond to a specific demand from the Lao PDR Ministry of Health for using locally available ingredients, but such a locally produced RUTF should not only be culturally acceptable but also effective in treating SAM in children [32,33,34]. Efforts are currently underway by a local, non-profit enterprise to produce mung-bean-based RUTF prototypes, which will be tested for effectiveness. Based on previous experiences and earlier acceptability studies in neighboring countries [16,19], it is expected that the locally produced Lao RUTF will be more acceptable than the currently used peanut-based RUTF, especially in the northern provinces of Lao PDR. Meanwhile, given the overall good acceptability, albeit not the preferred one, of the currently used peanut-based RUTF in the country, it is essential to sustain and keep up the efforts of the national program to fight acute malnutrition in Lao PDR by expanding IMAM coverage and implementation and improving the early detection of children with SAM.

## Figures and Tables

**Figure 1 nutrients-15-03847-f001:**
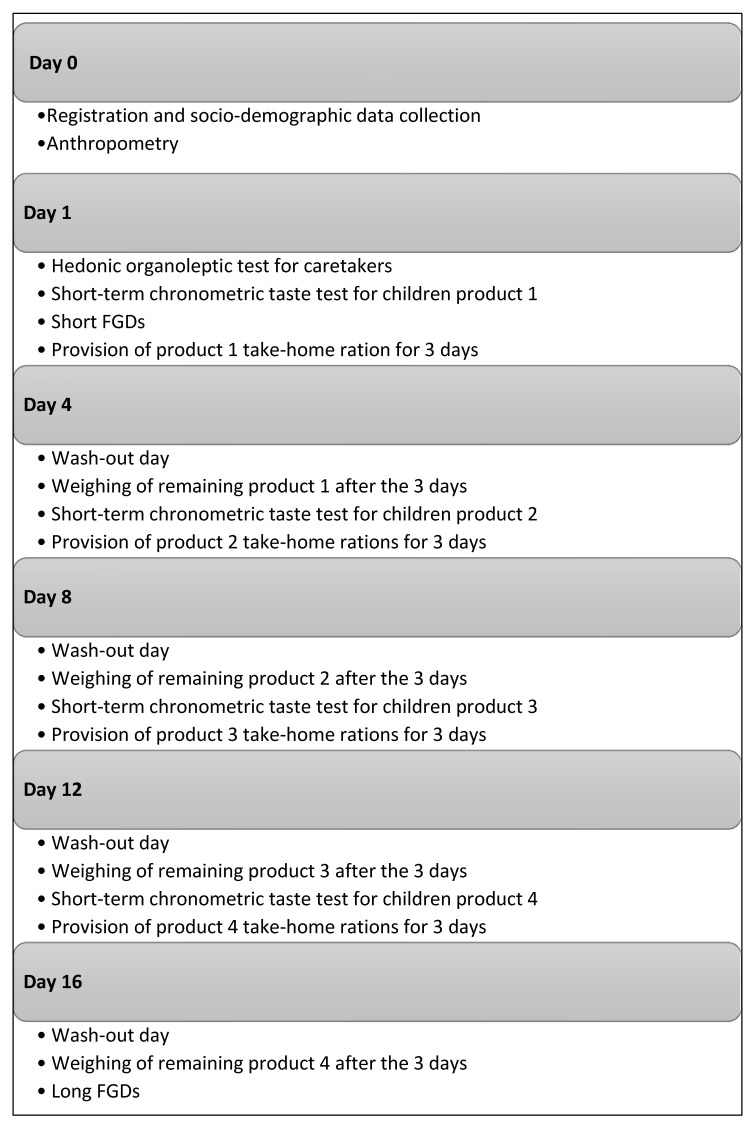
Study implementation scheme.

**Figure 2 nutrients-15-03847-f002:**
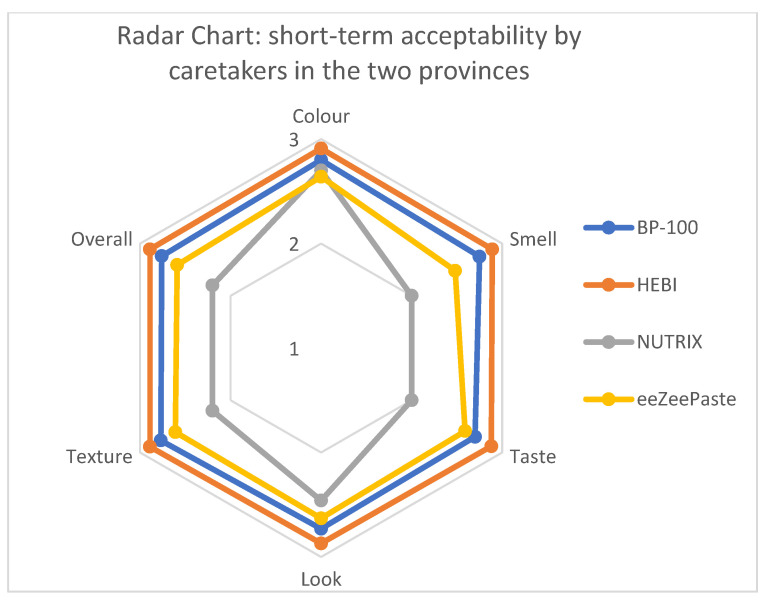
Short-term acceptability by caretakers.

**Table 1 nutrients-15-03847-t001:** Tested RUTF.

Name	Form	Ingredients	Origin
eeZeePaste	Paste	Peanut, milk	India
HEBI	Compressed cube	Mung bean, milk	Vietnam
Nutrix	Wafer + paste	Soy bean, rice, fish	Cambodia
BP-100	Compressed bar	Milk, wheat, oat	Norway

**Table 2 nutrients-15-03847-t002:** Characteristics of children.

	Phongsaly	Attapeu	Total	*p*-Value ^1^
n	41	42	83	-
Age, years (± SD)	2.76 ± 1.02	2.59 ± 1.24	2.67 ± 1.13	0.49
Sex, % male (n)	53.66 (22)	52.38 (22)	53.01 (44)	0.91
Weight, kg (± SD)	10.98 (± 1.68)	10.70 (± 2.09)	10.84 (± 1.90)	0.49
Height, cm (± SD)	83.16 (± 6.54)	85.75 (± 8.50)	83.96 (± 7.59)	0.34
MUAC, cm (± SD)	14.19 (± 0.97)	14.02 (± 0.90)	14.10 (± 0.93)	0.40
MUAC <= 12.5, % (n)	4.9 (2)	4.7 (2)	4.8 (4)	1
WHZ-SCORE (± SD)	−0.246 (± 0.866)	−0.952 (± 0.604)	−0.473 (± 0.852)	<0.01
Z-SCORE <= −2, % (n)	4.88 (2)	4.76 (2)	4.82 (4)	1

^1^ *p*-value tested with Student’s *t*-test, chi-square, Welch *t*-test, or Fisher’s exact test, depending on whether the conditions of application were met.

**Table 3 nutrients-15-03847-t003:** Mean composite ratings.

	Phongsaly				Attapeu				Total			
	BP-100	HEBI	Nutrix	eeZee P.	BP-100	HEBI	Nutrix	eeZee P.	BP-100	HEBI	Nutrix	eeZee P.
Colour	2.85	2.95	2.88	2.66	2.74	2.88	2.52	2.62	2.8	2.91	2.7	2.64
Smell	2.76	2.9	2.24	2.29	2.74	2.88	1.76	2.67	2.75	2.89	2	2.48
Taste	2.73	2.9	2.34	2.51	2.67	2.86	1.67	2.67	2.7	2.88	2	2.59
Look	2.76	2.9	2.56	2.59	2.71	2.83	2.36	2.67	2.73	2.87	2.46	2.63
Texture	2.78	2.93	2.41	2.56	2.76	2.86	2	2.67	2.77	2.89	2.2	2.61
Overall	2.76	2.93	2.37	2.54	2.76	2.86	2.05	2.64	2.76	2.89	2.2	2.59
Total (out of 18)	16.64	17.51	14.8	15.15	16.38	17.17	12.36	15.94	16.51	17.33	13.56	15.54
Acceptability %	92.44	97.28	82.22	84.17	91	95.39	68.67	88.56	91.72	96.28	75.33	86.33
Favorite product %	14.63	60.98	9.76	14.63	14.3	45.2	11.9	28.6	14.5	53	10.8	21.7
Worst product %	2.44	0	63.41	34.15	10	5	75	10	6.18	2.47	69.14	22.22

**Table 4 nutrients-15-03847-t004:** Intake of each RUTF over 3 days as a percentage of the total amount of RUTF provided.

RUTF	% of Intake over 3 Days
BP-100	92.4 ± 19.7
HEBI	93.4 ± 17.1
NUTRIX	81.8 ± 27.7
eeZeePaste	92.3 ± 20.9

## Data Availability

Requests for the original data should be addressed to the corresponding author.

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
