# Peer review of "Short-Term Acceptability of Ready-to-Use Therapeutic Foods in Two Provinces of Lao People’s Democratic Republic"

_nutrients, 2023, doi:10.3390/nu15173847_

Round 1

Reviewer 1 Report

This paper considers acceptability of four ready-to-use therapeutic foods (RUTF) tested in two study sites, Phongsaly and Attapeu.  The paper is straightforward and so I didn't have too much to say about the statistical methods used, other than I think some clarity is required. 

It is not always clear what statistical tests were used, and whether the ones chosen were the best to use.  For example, what was used to test the organoloptic scores.  Was it a t-test?  If it was, was the choice justified?  I am assuming that the distribution of scores is skewed and with a ceiling effect.  Perhaps a non-parametric test is preferred? I think clear statements around the choices of statistical methods should be included.

Was any considered given to the effect covariates may have on the outcomes?  Is this not of interest?

Table 4.  Heading states that these are mean percentages, but they are proportions.  Suggest to change to percentages as is presented in the text.

Paper was clearly written.

Author Response

This paper considers acceptability of four ready-to-use therapeutic foods (RUTF) tested in two study sites, Phongsaly and Attapeu.  The paper is straightforward and so I didn't have too much to say about the statistical methods used, other than I think some clarity is required.

It is not always clear what statistical tests were used, and whether the ones chosen were the best to use.  For example, what was used to test the organoleptic scores.  Was it a t-test?  If it was, was the choice justified?  

Reply: We would like to thank the reviewer for the time and efforts provided to review our manuscript. We have now specified in the manuscript which tests were used, and in the case of unequal variances, we used the Welch t-test.

I am assuming that the distribution of scores is skewed and with a ceiling effect.  Perhaps a non-parametric test is preferred? I think clear statements around the choices of statistical methods should be included.

Reply: We have added a statement accordingly (see line 191), and also added the type of statistical test performed before each and every p-value quoted in the manuscript.

Was any consideration given to the effect covariates may have on the outcomes?  Is this not of interest?

Reply: In our study independent variables were province, interviewer, caretaker-child relationship, caretakers’ age, caretakers’ gender, child age (DoB) and gender. Although quantitative analyses were performed both for all participants together, and separately for the two provinces, we did not dive deep into the effects that these covariates may have had on the outcomes, especially because our study used QUAN approaches to corroborate, compare, and cross-validate QUAL findings within the study (from FGDs), and because our sample size is not that large (n=83). In fact, this study mainly focused and put its attention on the Focus Group Discussion qualitative results: qualitative methods guided the project, and the quantitative ones provided a supporting role in the procedures.

Table 4.  Heading states that these are mean percentages, but they are proportions.  Suggest to change to percentages as is presented in the text.

 Reply: Thanks for pointing this out. We have corrected it.

Reviewer 2 Report

Introduction

Does Laos engage in micronutrients fortification of staple foods? Micronutrients in staple carbohydrates or  iodine in salt?

https://www.who.int/health-topics/food-fortification#tab=tab_1

Methods

FAO guidelines for running this type of trial appear to have been followed. 

Statistical methods section? 

Results

The established methods for assessing acceptability while robust and recognized, did not really differentiate between three products BP-100, HEBI and eeZeePaste

What about a Chi squared analysis on the percentage data in Table 4? Would there be significant differences in the % values indicating preferences between the RUTF products? As mentioned above;  I can see without doing a statistical test that the data for three of the products are not significantly different; the levels of aceptance across all brands appeared to be high; not surprising when the population was presumably the lowest income group? 

The focus group discussion material could be placed in the Appendix  and summarised in a paragraph in the results.

The acceptability percentages in Table 1 (actually Table2). and lines 214 - 216  indicate that care takers did not  discriminate in terms of acceptance between RUTF BP-100; HEBI and EeZeePaste ; quote  statistics  even though they are non significant. 

Section 3.2 in the results is the key point of the paper on acceptance by children and seems very brief. The replicate aspects of the data are not clearly explained. Average data are quoted with no indication of variation.

The data on RUTF  consumption after 30 min do not seem supported by data  in Table 4?

Each child  was presented with one of each form of RUTF paste over three days over the 'two to three' months of the trial.  There were 40 subjects in each location, so this section needs re-working in order to report the data in full.  Table 4  has no experimental replicate information and  indicates that all RUTF pastes became acceptable according to FAO standards of acceptability over the three days of presentation.

Discussion 

This rambles a lot and could be cut down by at least a third.

'What novel information is the study trying to add to what has already been established' is a good guideline to bear in mind for a Discussion

 The two population groups in this study are so low in income they can hardly feed themselves,  thus they are grateful for anything as beneficial as free nutrient balanced paste for young children. This was a confounding factor in the experimental design and should have been mentioned. 

Author Response

Introduction

Does Laos engage in micronutrients fortification of staple foods? Micronutrients in staple carbohydrates or iodine in salt?

Reply. We would like to thank the review for this time and efforts for reviewing our manuscript, which has enabled us to improve upon it.

Reply: The Government of Lao PDR has identified food fortification as a key intervention to improve the micronutrient intake of the population, under its National Nutrition Strategy to 2025. The Lao PDR Government has implemented mandatory salt iodization to prevent Iodine Deficiency Disorders (IDD) since 1995 (Decision No. 102 /MoH on Iodized Salt Standards).

More recently, WFP in Lao PDR is using fortified rice for its school meals programme, and working on local blending of fortified rice, hoping to introduce local fortification of rice. The Government of Lao PDR has identified both rice and cooking oil as foods for fortification (because of their widespread consumption).

Even though micronutrient deficiencies are part of the general term ‘malnutrition’, and are often highly prevalent in children with SAM,  we have not added more information on micronutrient fortification to the introduction as we feel it is outside the scope of the current manuscript, which focuses on acceptability of RUTFs and SAM treatment.

Methods

FAO guidelines for running this type of trial appear to have been followed.

Statistical methods section?

Reply: we have added a clear statement (see line 191), and also added the type of statistical test performed before each and every p-value quoted in the manuscript).

Results

The established methods for assessing acceptability while robust and recognized, did not really differentiate between three products BP-100, HEBI and eeZeePaste.

Reply: correct. Indeed, the manuscript ends by saying that given the overall acceptability of the currently used RUTF in the country (eeZeePaste), albeit not the preferred one, it is essential to sustain and keep up the efforts of the national program to fight acute malnutrition in Lao PDR and continue treating children with the current RUTF. After 30 minutes, all the RUTF tested had been eaten on average more than their 50% of the initial amount, but no statistically significant difference was found among the four RUTF (ANOVA p-value>0.05).

However, it is perhaps also important to refer to a recent trial in Indonesia, which reported that although peanut-based RUTF was found acceptable in short trials, consumption over a 8 week period was 50% lower than for a mung-based RUTF.

What about a Chi squared analysis on the percentage data in Table 4? Would there be significant differences in the % values indicating preferences between the RUTF products? As mentioned above;  I can see without doing a statistical test that the data for three of the products are not significantly different; the levels of acceptance across all brands appeared to be high; not surprising when the population was presumably the lowest income group?

Reply: Indeed, differences in acceptability were small for 3 of the 4 RUTFs, and even with a much larger sample size, we doubt whether these differences would have reached statistical significance. We did perform statistical analysis on data presented in Table 4: the only statistically significant difference is found between BP-100 and Nutrix (Welch t-test p-value 0.007), and between HEBI and NUTRIX (Welch t-test p-value 0.002), but not statistically significant difference between the other three (HEBI, BP-100, Eezypaste), as already assumed by the reviewer. Besides statistical significance, we feel there is also a ‘practical significance’ in that our results show that for the moment, there are no strong reasons to switch to another RUTF and the Lao government can focus on implementing the IMAM program, while the locally produced RUTF is further developed.

We selected poor areas in both provinces for the study. To recruit children-caretakers’ pairs on site, contact was sought with primary health centers near the villages of Muang Mai (Phongsaly) and Sanamxai (Attapeu). Thanks to the help of the health centers, the village chief, and the municipality, caretakers and their children where gathered at the health post or village hall, where the study was explained, and participation requested. Thus, the recruitment has been carried out with the “convenience sampling” technique, but indeed represent a part of the Lao population with lower SEC.  

 The focus group discussion material could be placed in the Appendix  and summarised in a paragraph in the results.

Reply: We have reduced the amount of text in the discussion concerning the FDGs considerable.

The acceptability percentages in Table 1 (actually Table2). and lines 214 - 216  indicate that care takers did not  discriminate in terms of acceptance between RUTF BP-100; HEBI and EeZeePaste ; quote  statistics even though they are non significant.

Reply. Thank you for pointing this out. We have added relevant statistics accordingly. Also, as for the tests used, note 1 below Table2 was adapted to better explain the details.

Section 3.2 in the results is the key point of the paper on acceptance by children and seems very brief. The replicate aspects of the data are not clearly explained. Average data are quoted with no indication of variation.

Reply: We have added details accordingly (see paragraph 3.2).

The data on RUTF consumption after 30 min do not seem supported by data in Table 4?

Reply: Indeed, preference by caretakers and intake by children were not 100% in accordance. Especially in Phongsaly (North Laos), Eezypaste was in 1/3 of the cases picked as the ‘worst’ RUTF, which is worrisome. However, consumption by the children was not notable lower.
Caretakers and children did tend to agree on that HEBI, the mung bean based RUTF was the best RUTF and Nutrix the worst. We have tried to point out more clearly these differences in the manuscript now. Please see the first paragraph of the discussion for example.

Each child was presented with one of each form of RUTF over three days over the 'two to three' months of the trial.

Reply: The study implementation was 16 days in total (see Figure 1).

There were 40 subjects in each location, so this section needs re-working in order to report the data in full. Table 4 has no experimental replicate information and indicates that all RUTF pastes became acceptable according to FAO standards of acceptability over the three days of presentation.

Reply. Indeed, over the 3 days consumption period for each RUTF, all RUTF fulfilled the FAO criteria. Therefore, the article ends by saying that given the overall good acceptability of the four tested RUTF (and generally with no statistically significance difference among them), it is essential to sustain and keep up the efforts of the national program to fight acute malnutrition in Lao PDR and continue treating children with the current RUTF.

However, we also note the shortcomings of the present study, especially that the consumption was only for 3 days and not 6 to 8 weeks, as in a real treatment program. The Ministry of Health of Lao PDR program managers raised the concern that the currently used RUTF was potentially not acceptable and expressed a concern on using imported products. Our study partly confirms this (high prevalence of ‘worst product’ in Phongsaly) but also shows that consumption appears less affected, and hence, there is no direct need to shift to another RUTF

 Discussion

This rambles a lot and could be cut down by at least a third.

Reply: We have completely revised the discussion

'What novel information is the study trying to add to what has already been established' is a good guideline to bear in mind for a Discussion

The two population groups in this study are so low in income they can hardly feed themselves,  thus they are grateful for anything as beneficial as free nutrient balanced paste for young children. This was a confounding factor in the experimental design and should have been mentioned.

Reply: please see the revised discussion.